# Wideband Omnidirectional Antenna Featuring Small Azimuthal Gain Variation

**DOI:** 10.3390/mi14122218

**Published:** 2023-12-08

**Authors:** Honglin Zhang, Zhenzhan Fu, Binjie Hu, Zhijian Chen, Shaowei Liao, Bing Li

**Affiliations:** 1School of Electronic and Information Engineering, South China University of Technology, No. 381 Wushan Road, Guangzhou 510641, China; eebjiehu@scut.edu.cn (B.H.); liaoshaowei@scut.edu.cn (S.L.); 2School of Microelectronics, South China University of Technology, No. 381 Wushan Road, Guangzhou 510641, China; 18370816621@163.com (Z.F.); chenzhijian@scut.edu.cn (Z.C.); 3School of Electrical Engineering, Southwest Jiaotong University, Chengdu 611756, China; bl@swjtu.edu.cn; 4State Key Laboratory of Millimeter Waves, Southeast University, Nanjing 210096, China

**Keywords:** omnidirectional antenna, wideband, impedance matching, pattern superposition

## Abstract

Wideband omnidirectional antennas are essential components in radio monitoring and communication systems, enabling the reception of signals from all directions over a wide bandwidth. This paper presents a novel wideband omnidirectional antenna design that achieves a 1-dB gain variation across its azimuthal plane within a bandwidth of 1.8 GHz to 7.77 GHz. The antenna’s exceptional performance is attributed to two flower-bud-shaped monopoles that, through pattern superposition, generate a wideband omnidirectional radiation pattern. Analysis shows that the use of a circular ground plane also reduces the azimuthal gain variation. Additionally, an embedded matching structure integrated into the antenna’s base enhances the impedance bandwidth without compromising its compact size. Analytical investigations demonstrate that the matching structure effectively behaves as a five-order LC circuit, explaining its wideband matching capabilities. Furthermore, structural modifications effectively reduce side lobe levels, ensuring minimal interference. Experimental measurements corroborate the antenna’s omnidirectional radiation pattern and confirm that the azimuthal gain variation remains within 1-dB throughout its bandwidth, while maintaining an S11 below −10 dB from 1.8 GHz to 7.7 GHz. The antenna’s bandwidth overlaps with the spectrum intensively used in mobile communication technologies, such as LTE, Bluetooth, and IEEE 802.11be, as well as radiolocation applications, making it a promising choice for unmanned aerial vehicles conducting communication and radio monitoring missions.

## 1. Introduction

In communication, radar, and radio monitoring systems, wideband omnidirectional antennas play a critical role. These antennas receive signals from all directions over a broad bandwidth, making them indispensable components for a wide range of applications, including communication, television, and radio monitoring [1,2]. The omnidirectional antennas featured in [1,2] possessed bandwidths exceeding 40%, enabling signal reception from any direction in the azimuth plane. Researchers have continuously sought to expand antenna bandwidth, leading to innovative designs that push the boundaries of performance. One such advancement is the extremely thin dipole driven by a vertical balun introduced in [3]. This design achieved a remarkable bandwidth of 91.2% while maintaining a simple and lightweight structure. For radio monitoring and high-throughput communication systems, antennas with bandwidths exceeding 100% of their center frequency have become increasingly common [4,5,6]. A prime example is the biconical antenna fed by a novel balun presented in [7], which achieved a bandwidth spanning from 3 GHz to 20 GHz. Its radiation patterns exhibited minimal side lobes at the higher end of the bandwidth on the elevation plane.

Further bandwidth enhancements were achieved through the application of two coupled loops with capacitive feed, as described in [8]. This technique resulted in a significantly wider bandwidth, reaching 122% while preserving a compact antenna size. The Vivaldi structure, introduced in [9], yielded a bandwidth of 123%, while a CPW feeding line was employed in [10]. In [11], a monopole with three parallel strips specifically designed for backpack applications produced a bandwidth of 124.8% and a lightweight structure. Additionally, an inverted cone monopole with a wide bandwidth ranging from 1 to 4 GHz was presented in [12].

To address the horizontal gain variation issue, ref. [13] proposed the use of multiple radiators instead of a single radiator. The antennas in [3,4,5,6,7,8,9,10] possessed bandwidths sufficient to cover the fundamental and higher order modes, leading to significant gain variation on the azimuthal plane.

Beyond bandwidth, antenna weight is a crucial consideration for airborne applications because of payload limitations [6,14]. To address this challenge, ref. [15] introduced a substrate integrated dielectric resonator antenna with a bandwidth of 49.2% and a profile of 0.1 wavelength at the center frequency. A low-profile antenna utilizing a 1-to-8 equal output divider to feed a circular patch was presented in [16], achieving a bandwidth from 1.57 GHz to 5.39 GHz.

This paper presents a novel wideband omnidirectional antenna design that offers exceptional performance in terms of bandwidth, gain variation, size, and weight. The antenna’s simple structure comprises two printed monopoles that combine to produce a circular radiation pattern. Embedded matching patches and open-end strips enhance the antenna’s matching and radiating characteristics, ensuring optimal performance across its wide bandwidth. The proposed antenna’s exceptional features make it an ideal candidate for both land-based and airborne communication and radio monitoring applications.

## 2. Materials and Methods

Figure 1 shows the configuration and dimensions of the proposed antenna. As depicted in the three-dimensional view in Figure 1a, two patches, reminiscent of flower buds, are printed on a pair of laminates that are orthogonally positioned on a circular ground plane. Each laminate is printed with identical metal patterns on both surfaces, interconnected through a series of metallic via holes. The metal patterns and their corresponding dimensions are illustrated in Figure 1b. The flower-bud-shaped radiating structure is excited by a metal strip connected to an SMA connector on the back of the ground plane.

The primary radiating structure is the flower-bud-shaped patch, which features a horizontal θ-shaped structure and four open-end stubs. To facilitate wideband matching, two right-angled pentagons are placed at the base of the feed line. Five dimensions (*W*_1_, *W*_2_, *W*_3_, *H*_2_, and *H*_3_) control impedance matching. Table 1 lists the fine-tuned dimensions of the printed patch patterns.

Figure 2 illustrates the design process of the proposed antenna. Figure 2a,b shows the initial monopole and its crosswise combination. Figure 2c,d depicts the addition and modification of the embedded matching structure. Figure 2e displays the last design with open stubs for side lobe reduction.

Simulations reveal that the radiation pattern of the monopole (Antenna I) in Figure 2a changes as the frequency increases from 2 GHz to 8 GHz. At the lower end of the band, the monopole radiates omni-directionally. However, at frequencies above 4 GHz, the radiation pattern on the azimuthal plane (xoy-plane) becomes an ellipse. Using the superposition principle [17,18], a circular pattern is generated on the xoy-plane by positioning two identical Antenna Is (denoted by Ant. *A* and Ant. *B*), depicted in Figure 2a orthogonally, as shown in Figure 3. Figure 3a shows the side views of the combination process and the corresponding radiation patterns on the yoz-plane. The top views of the combination stages and their relating patterns on the xoy-plane are shown in Figure 3b, which shows that the vertical- and horizontal-oriented elliptical patterns of Ant. *A* and Ant. *B* are added together to form a circle. Figure shows that the superposition operation keeps the radiation pattern on the yoz-plane but creates an omnidirectional pattern on the xoy-plane by superimposing two perpendicularly oriented elliptical patterns. By using the superposition principle, Antenna II is designed by positioning two identical Antenna Is crosswise over the ground plane.

## 3. Results

Simulation results are obtained with the CST Microwave Studio and analyzed to show the mechanisms of matching and radiation pattern control.

### 3.1. Bandwidth Expansion and Radiation Pattern Control Mechanisms and Simulation Results

Wideband Impedance Matching Mechanism

Wideband matching can be achieved by adding two matching patches at the base of the feed line. Antenna III is designed by adding two rectangular patches to Antenna II, as shown in Figure 2c. The patches significantly change the impedance values from 2 GHz to 4.6 GHz and lead to a broad bandwidth from 2.6 GHz to 6.9 GHz, as shown by the impedance and *S*-parameter curves in Figure 4.

The matching patches’ role has been analyzed to reveal its critical role in improving the bandwidth. Figure 5a shows the side view of Antenna III and three key dimensions relating to bandwidth enhancement.

Figure 5b illustrates that increasing $W_{1}$, the width of the patches, improves the matching between 5 GHz and 8 GHz. The matching between 2 GHz and 8 GHz improves with a larger value of $W_{2}$, which is the distance between the feed line and the inner vertical edges of the rectangular patches, as shown in Figure 5c. Figure 5d demonstrates that the matching between 2.5 and 6 GHz is much better when the patches height $H_{2}$ is 7 mm. Figure 5 suggests the bandwidth could be increased by choosing certain values of $W_{1}$, $W_{2}$, and $H_{2}$.

The matching structure in Antenna III can be treated as an embedded circuit that provides impedance matching in a wide bandwidth. Because the matching happens after adding the rectangular patches, the circuit construction should reflect the matching structure. An L-C circuit is constructed based on the assumption that the feed line conducts well and adds some inductance between the horizontal θ-shaped structure at high frequencies, and the assumption that parallel capacitance exists between the feed line and the matching patches because of the electrical field between the feed line and the matching patches shown in Figure 6a. The figure shows the electrical field leaving from the feed line and ending at the edges of the matching patches that resemble a capacitor. Considering the inductance and capacitance are distributed along the feed line, and a circuit with only an inductor and a capacitor can not match the antenna in a wide bandwidth, multiple inductors and capacitors are used to construct the circuit. The order and elements’ values of the circuit are obtained through an optimization method similar to those introduced in [19,20,21]. To decide the order of the L-C circuit, a guess and try method is applied to find the smallest order of the circuit to mimic the matching structure. Simulation shows that a five element L-C circuit shown in Figure 6b can represent the matching structure.

The circuit’s element values are determined by forcing $Z_{in}^{\prime }=Z_{in}$ in the circuit simulation tool. To do that, impedance $Z_{A}$ and $Z_{in}$ are extracted from the *S*-parameters with an excitation port at reference line AA’ and BB’, respectively. Because $Z_{A}$ is obtained at reference line BB’, it can be treated as the input impedance of the radiation structure. Another set of impedance $Z_{in}^{\prime }$ is obtained by simulating the circuit in Figure 6b. Use optimization algorithms provided by the circuit simulator (Advanced Design System) and use $Z_{in}$ as the optimization goal, comparing $Z_{in}^{\prime }$ with $Z_{in}$ at each end of the iterations, the circuit’s elements values are decided. When $Z_{in}^{\prime }=Z_{in}$, the five-order circuit is equivalent to the matching structure in the dashed box shown in Figure 6c. In this case, the following relation holds
(1)$$Z_{in}^{\prime }=\frac{i\omega \left(-C_{2}L_{2}L_{3}\omega^{2}+L_{2}+L_{3}\right)-C_{2}L_{2}\omega^{2}Z_{A}+Z_{A}}{C_{1}C_{2}L_{2}L_{3}\omega^{4}-\omega^{2}\left(C_{1}\left(L_{2}+L_{3}\right)+C_{2}L_{3}\right)-iC_{1}C_{2}L_{2}\omega^{3}Z_{A}+i\omega Z_{A}\left(C_{1}+C_{2}\right)+1}+iL_{1}\omega$$

By finding exact values of the elements in (1), the input impedance $Z_{in}^{\prime }$ of the equivalent circuit could be the same as $Z_{in}$ of the 3D model. The simulation shows the matching can be controlled by tuning the height and length of the rectangular patches and their distance to the feed line. When the dimension values are set, the inductors’ and capacitors’ values are determined through calculation.

Figure 7 shows the simulated reflection coefficients of Antenna III obtained from its 3D model and equivalent circuit. It shows the reflection coefficient of the equivalent circuit is nearly identical to the result of the 3D model. The agreement verifies that the five-order circuit is equivalent to the matching structure.

Side lobe Suppression

However, adding matching patches to Antenna II produces large side lobes, as shown in Figure 8, which shows the radiation patterns of Antenna II and Antenna III. The only difference between the two antennas is that Antenna III includes the rectangular matching patches. In Figure 8a, two side lobes around θ=0° only exist at 7 GHz. However, four side lobes with larger values appear at 7 and 8 GHz in the patterns of Antenna III. Because the only difference between the two antennas is the addition of the matching patches in Antenna III, it suggests the matching patches in Antenna III cause the large side lobes.

Figure 9 shows the differences between the electrical distributions of the two antennas. Two major differences exist, as indicated by the large red arrows. The first difference is that the strong field leaving the θ-shaped structure reaches the matching patches’ horizontal edges in Antenna III rather than reaching the ground plane. The second difference is that the strong field leaves the ground plane and reaches the vertical edges of the matching patches in Antenna III. These differences suggest both the horizontal and vertical edges redistribute the field and cause the large side lobes in the radiation patterns of Antenna III.

Simulations show that cutting the outer corners of the matching patches, resulting in irregular pentagons, as illustrated in Figure 10a, can restore the field paths, as indicated by the large red arrows. It is also observed that the field leaves the longest edge of the pentagons and reaches the θ-shaped structure. Because of the cutting, the side lobe levels are significantly reduced, as shown in Figure 10b,c. The radiation patterns of Antenna III and Antenna IV reveal that the new configuration reduces the side lobes by 6.5 dB and 7 dB at 7 and 8 GHz, respectively.

However, the side lobes of Antenna IV at 7 and 8 GHz are still large according to Figure 10c. It shows the side lobe levels at the two frequencies are only 6 to 7 dB smaller than the main lobe levels around θ=15° and θ=345°.

Simulation attempts suggest that adding open stubs can change the field distribution and reduce the side lobe levels. Figure 11 demonstrates the tree open-stub configurations, their radiation patterns, and relevant field distributions. Simulation results suggest the third configuration in Figure 11c produces wider beamwidth and no side lobes near θ=0°. Figure 11 shows the structure in Figure 11a produces two large side lobes around θ=20° and θ=340°, in which the structure in Figure 11b significantly reduces the side lobes as indicated by Figure 11e. Type 3, shown in Figure 11c, is designed by adding thin slits to the open stubs in Figure 11b. This configuration further reduces the side lobe levels, as shown by the patterns in Figure 11f.

Because the radiation patterns in Figure 11f have no side lobes at 7 and 8 GHz, the corresponding configuration is selected and fine-tuned for prototyping and testing. The optimized 3D model is also analyzed with the five-order LC circuit shown in Figure 6b. Figure 12 demonstrates the side view of the proposed antenna, a five-order circuit loaded with the impedance at BB’, and corresponding reflection coefficient curves. The agreement of the reflection curves in Figure 12c proves the equivalent circuit correctly mimics the matching structure between AA’ and BB’ shown in Figure 12a.

Current distribution of the antenna

Because the antenna works in a wide bandwidth, the current distribution on its structure surface gives insightful information on how the antenna works. Figure 13, Figure 14 and Figure 15 show the distributions at different phases at 2, 5, and 7 GHz, which are located in the lower, intermediate, and upper regions of the bandwidth.

The current distributions show strong current exists on most of the antenna’s radiating and matching structure when the phases are 45, 90, and 135 degrees at 2 GHz, as shown by Figure 13b–d. It suggests the horizontal θ-shaped structure and the open stubs play important roles at 2 GHz.

However, the current intensity on the horizontal θ-shaped structure reduced significantly at 5 and 7 GHz, as shown in Figure 14 and Figure 15. Figure 14 also shows that the current is less dense at 7 GHz on the open stubs near the horizontal θ-shaped structure. The current distributions in Figure 13, Figure 14 and Figure 15 suggest the outer open stubs play an important role at all three frequencies. In addition, the figures also show the current on the matching structure is strong at the three frequencies, meaning the structure is critical for the antenna.

Ground shape and size choice.

The antenna’s ground impacts the antenna’s performances regarding its azimuthal gain variation and bandwidth.

In the design process, pattern comparison has been conducted to select the shape of the ground plane for prototyping and testing. For fair comparison, all the ground planes in Figure 16 have the same area of 17,671.5 mm^2^, which is the area of the circular ground planes with a diameter of 150 mm. The correspondence parameters of the ground planes are
(2)$$d_{circle}=2\sqrt{\frac{S}{\pi }},$$
(3)$$a_{s}=\frac{\sqrt{\pi }}{2}d_{circle},$$
(4)$$a_{t}=\frac{2\sqrt{S}}{\sqrt[{4}]{3}}=\frac{\sqrt{\pi }}{\sqrt[{4}]{3}}d_{circle}.$$
where $d_{circle}$ is the diameters of the circle and $a_{s}$ and $a_{t}$ are the side lengths of the square and equilateral triangle. To obtain the same area, the diameters of the circumscribed circles of the square and equilateral triangle are
(5)$$d_{c,s}=\sqrt{\frac{\pi }{2}}d_{circle}\approx 1.25d_{circle},$$
(6)$$d_{c,t}=4\frac{\sqrt{S}}{\sqrt[{4}]{27}}=\frac{2\sqrt{\pi }}{\sqrt[{4}]{27}}d_{circle}\approx 1.56d_{circle}.$$
where $d_{c,s}$ and $d_{c,t}$ are the diameters of the circumscribed circles. Equations (5) and (6) show that the diameter of a circular plane is smaller than the diameters of the circumscribed circles of equilateral triangular and square ground planes.

Figure 17 shows the radiation patterns obtained with equilateral triangular, square, and circular ground planes at 2, 5, and 7 GHz. The patterns in Figure 17a show that the triangular-shaped ground plane produces the largest gain variation at 2 GHz, where the variation is over 4 dB. The gain variation gets smaller but with more ripples as the frequency increases for the antenna with a triangular ground plane. Similar phenomena are observed regarding the patterns of the antenna with a square ground plane. Contrary to the cases with triangular and square ground planes, the gain variations at all three frequencies are negligible.

Based on the comparison of radiation pattern, a circular ground plane is selected.

Our analysis shows that a larger diameter leads to a wider bandwidth, as demonstrated by the reflection curves shown in Figure 18. The five $S_{11}$ curves in the figure are obtained with different diameters of the circular ground plane.

The curves in Figure 18 show that increasing the diameter from 120 mm to 180 mm increases the bandwidth by including more matched frequencies at the lower end of the bandwidth. When $d_{circle}$ is increased from 120 mm to 180 mm, the lower boundary of the bandwidth goes from 2.34 GHz to 1.68 GHz, increasing the bandwidth by 660 MHz. However, the ground plane’s diameter can not be increased indefinitely because the simulation shows that the larger ground plane also produces larger side lobes around the zenith axis, as shown Figure 19.

The curves in Figure 19 show that the side lobes near θ = 0 become larger as the diameter increases. Considering that the larger ground plane offers wider bandwidth but also increases side lobe levels, an intermediate value should be chosen for the ground plane’s diameter. At the end of the design, the diameter should be 150 mm.

### 3.2. Time Domain Analysis

The proposed antenna is analyzed in the CST Microwave Studio to show its time domain performance following the analysis set-ups discussed in [22,23,24,25]. Figure 20a shows the simulation configuration in the simulation. Two identical antennas are placed at a distance of 800 mm, which is about 5 wavelengths at 1.8 GHz.

Time analysis uses Gaussian pulse as the excitation signal for the two antennas [24]. The simulation result in Figure 12b shows the group delay between the two antennas is below 4 ns in the antenna’s bandwidth from 1.8 GHz to 7.77 GHz.

### 3.3. Measurement Results and Discussion

The antenna is fabricated and tested for verification. Figure 21 shows the fabricated prototype of the antenna, which is printed on Rogers Corp.’s 4003C laminates with a thickness of 0.508 mm, relative permittivity of 3.55, and tangential loss of 0.0027. The metal patterns are printed on both sides of the laminates and connected with metallic via holes. Necessary cutouts are made to position the laminates as desired. The ground plane is chosen to be a circular patch with a diameter of 150 mm, which is a tradeoff between desired radiation and size.

The measured *S*-parameter curve agrees well with the simulated one, as shown in Figure 22. The measured data are obtained with Keysight’s E5071C VNA. The measured *S*_11_ shows the antenna’s impedance bandwidth is from 1.80 GHz to 7.77 GHz, corresponding to a fractional bandwidth of 124.8%.

The antenna’s radiation performance is tested from 1 to 6 GHz because the testing system (Microwave Vision Group’s StarLab) can only cover this frequency band.

Figure 23 plots the realized gain and radiation efficiency of the antenna. The gain is from 1.24 to 5.44 dBi and from 1.8 GHz to 6 GHz, which agrees well with the simulated result. The measured bandwidth is slightly wider than the simulated one. In the simulation, the lowest and highest frequencies of the bandwidth are 1.89 GHz and 7.68 GHz. The lowest and highest frequencies of the measured band are 1.8 GHz and 7.77 GHz in the measurement. In addition, the matching measured reflection is smaller around 2.6 GHz and between 4.1 GHz and 6.2 GHz. The discrepancies may come from the difference between the model and the fabricated prototype. Other than that, the measured reflection is very close to the simulated one. It shows the measured efficiency is larger than 70% but is smaller than the simulated result. The measured efficiency is smaller than the simulated efficiency below 5.9 GHz. A smaller measured efficiency is reasonable because there may be some difference between the fabricated prototype and the simulated model. In addition, the metal parts of the antenna are printed on both sides of the substrates, connected by metalized via holes. These configurations may lead to discrepancies between the measured and simulated results. Besides that, there is a sudden rise in the measured efficiency at 6 GHz. Because it is located at the upper limit of the testing system, we think the testing system may need upgrading to solve this problem. Figure 23 also shows the measured gain changes in the same way as the simulated gain does, but with slightly smaller values in the bandwidth. Because the measured efficiency is approximately 15% to 26% smaller than the simulated one, smaller measured gain is expected. However, the measured gain values are larger than the simulated ones at 2.6 GHz, 3.4 GHz, 3.7 GHz, and frequencies beyond 5.3 GHz. Since the difference is small and the measured gain curve has the same tendency as that of the simulated one, the measured data verify the simulation.

Figure 24 shows the measured and simulated radiation patterns of the antenna at 2, 4, and 5.5 GHz, respectively. It shows that the simulated and measured patterns on the xoy- and yoz-plane agree well with each other at the frequencies. The measured cross-polarization levels at the three frequencies are smaller than −25 dB, while the simulated cross-polarization radiation level is below −30 dB and invisible on the xoy- and yoz-plane. The gain variation on the xoy-plane at each frequency is measured, and it is smaller than 1 dB according to the measured data. Simulation shows the gain variation between 6 GHz and 7.7 GHz is less than 1 dB. The agreement between measurement and simulation at 2, 4, and 5.5 GHz shown in Figure 16 confirms the proposed antenna has small gain variation on the xoy-plane at these frequencies.

## 4. Discussion

In this section, we compare the proposed antenna with other wideband omnidirectional antennas in terms of the structure type, bandwidth, gain, azimuthal gain variation, and size (Table 2). While the proposed antenna may not be the smallest or the highest-gain antenna in the table, it offers several distinct advantages that make it a compelling choice for a wide range of applications.

The proposed antenna exhibits a remarkable fractional bandwidth of 124.8%, closely approaching the widest fractional bandwidth of 127.27% reported in [16]. The wide bandwidth allows for seamless coverage of a broad spectrum of frequencies, making it suitable for applications that require comprehensive signal reception in a very wide bandwidth.

Significantly, the proposed antenna distinguishes itself with its small gain variation, maintaining a remarkable 1-dB variation throughout its bandwidth. This minimal gain variation ensures a consistent signal receiving or transmitting ability across all directions, making it ideal for applications where signal uniformity is crucial.

Moreover, the proposed antenna’s lightweight construction, enabled by its fabrication using printed circuit board (PCB) laminates, makes it an excellent choice for airborne applications. Its weight can be further reduced by using a thin substrate plated with copper sheets, interconnected by metallic via holes, on both sides. Its compact size and minimal weight allow for easy integration into unmanned aerial vehicles (UAVs), enabling them to carry additional payloads without compromising their performance.

In conclusion, while the proposed antenna may not be the absolute best in terms of size or gain, its combination of wide bandwidth, small gain variation, and lightweight construction makes it a compelling choice for various applications, particularly in the realm of airborne communication, radio monitoring, and radiolocation.

## 5. Conclusions

In conclusion, this paper presents a novel wideband omnidirectional antenna design that exhibits 1-dB gain variation across its azimuthal plane within a bandwidth of 1.8 GHz to 7.77 GHz. Additionally, the antenna’s simple structure, comprising two printed circuit boards and a circular ground plane, facilitates easy fabrication and integration into various systems. Extensive simulation analyses and tests have validated the proposed antenna’s superior performance, confirming that it radiates omni-directionally with minimal gain variation within its operational bandwidth while maintaining an S11 below −10 dB from 1.8 GHz to 7.7 GHz. These results underscore the antenna’s suitability for a wide range of communication and monitoring applications, including mobile communication, radiolocation, and electromagnetic environment monitoring on unmanned vehicles and airplanes.

## Figures and Tables

**Figure 1 micromachines-14-02218-f001:**
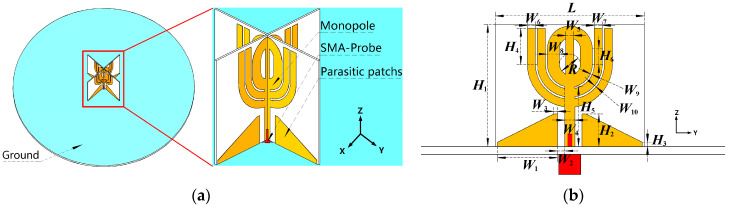
Geometry of the proposed wideband omnidirectional antenna with bud-shaped structure. (**a**) 3D view of the proposed antenna. Light blue, white, and yellow colors represent the ground plane, the substrate, and the metallic patterns of the antenna, respectively. (**b**) Detailed dimensions of the proposed antenna. Yellow and red colors represent the metallic patterns and the SMA connector of the antenna, respectively.

**Figure 2 micromachines-14-02218-f002:**
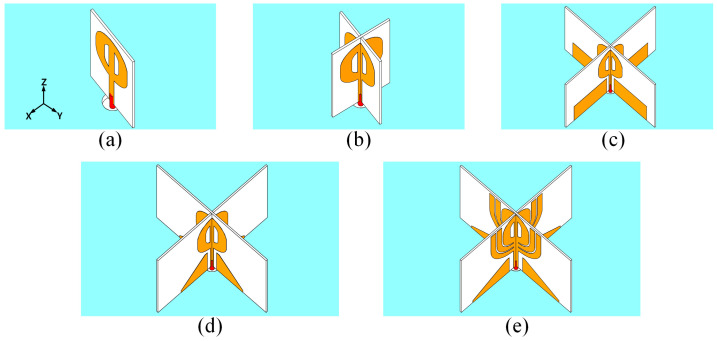
The design process of the proposed antenna. (**a**) Monopole antenna with inverted *θ* structure (Antenna I). (**b**) Vertically crossed monopole antenna with horizontal θ-shaped radiating structure (Antenna II). (**c**) Vertically crossed monopole antenna with rectangular patches (Antenna III). (**d**) Vertically crossed monopole antenna with right-angled pentagonal patches (Antenna IV). (**e**) Final design utilizing a pair of open stubs for improved radiation pattern (Antenna V).

**Figure 3 micromachines-14-02218-f003:**
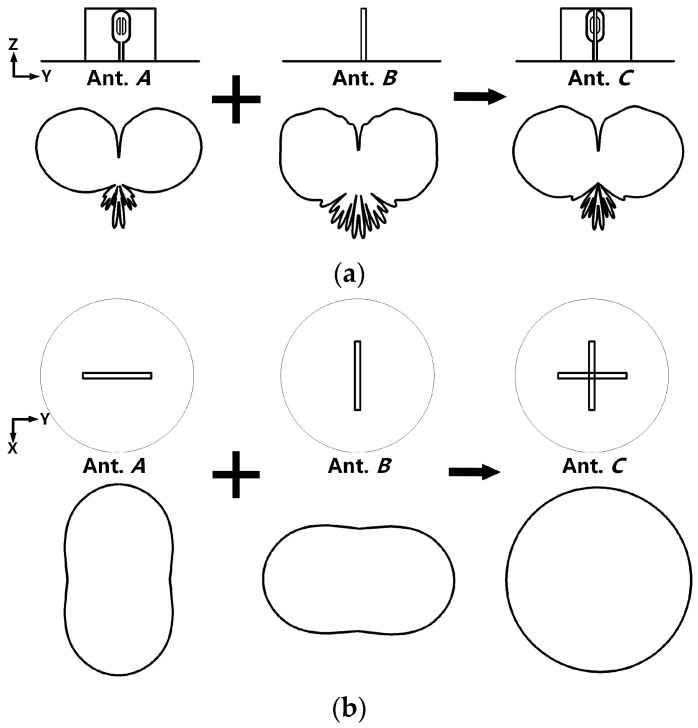
Combination process guided by the superposition principle. (**a**) Side views of the combined antennas and their radiation patterns on yoz-plane. (**b**) Side views of the combined antennas and their radiation patterns on xoy-plane.

**Figure 4 micromachines-14-02218-f004:**
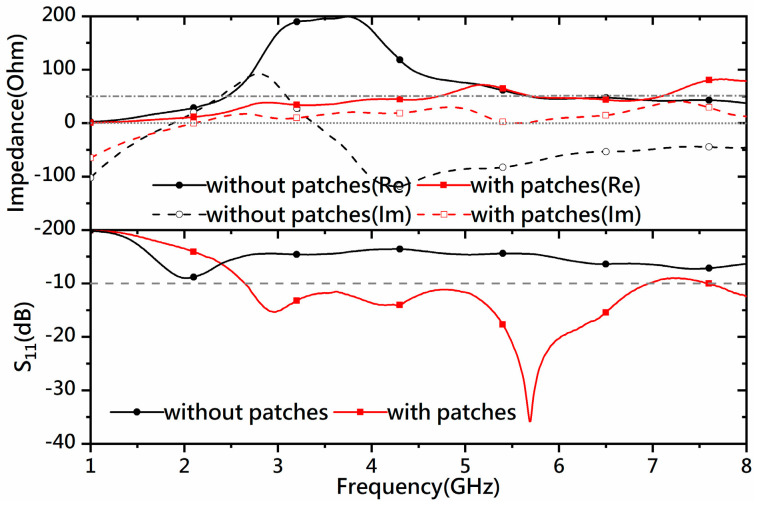
Simulated impedances and reflections (*S*_11_) before and after adding the rectangular matching patches.

**Figure 5 micromachines-14-02218-f005:**
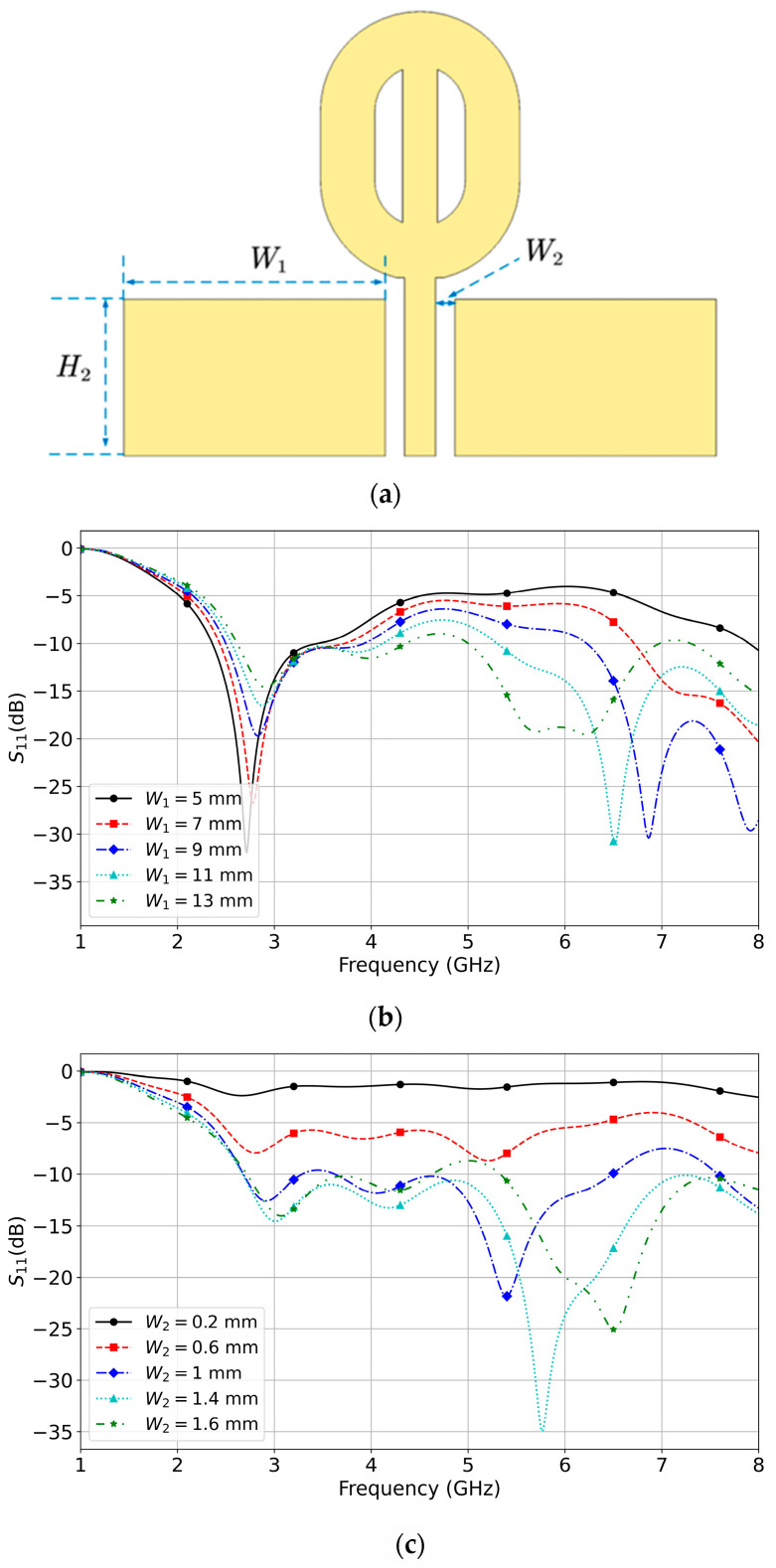
Antenna III and parametric study results. (**a**) Side view of Antenna III. (**b**) Parametric study results regarding $W_{1}$. (**c**) Parametric results regarding $W_{2}$. (**d**) Parametric study results regarding $H_{2}$.

**Figure 6 micromachines-14-02218-f006:**
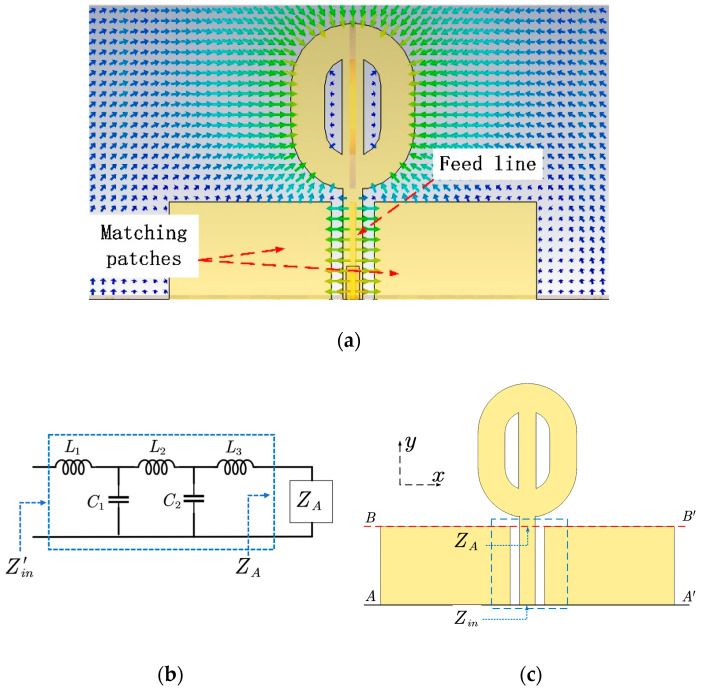
Field distribution, equivalent circuit model, and side view of Antenna III. (**a**) Side view and electrical field distribution of Antenna III. (**b**) Equivalent circuit model of the matching structure loaded with *Z_A_*. (**c**) Side view of Antenna III with different reference lines used for impedance extraction.

**Figure 7 micromachines-14-02218-f007:**
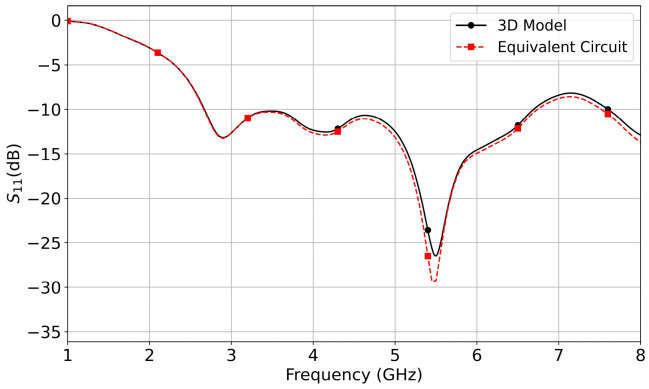
Simulated reflection coefficients of Antenna III. The corresponding element values are *L*_1_ = 0.427 nH, *L*_2_ = 0.824 nH, *L*_3_ = 0.261 nH, *C*_1_ = 0.289 pF, and *C*_2_ = 0.237 pF.

**Figure 8 micromachines-14-02218-f008:**
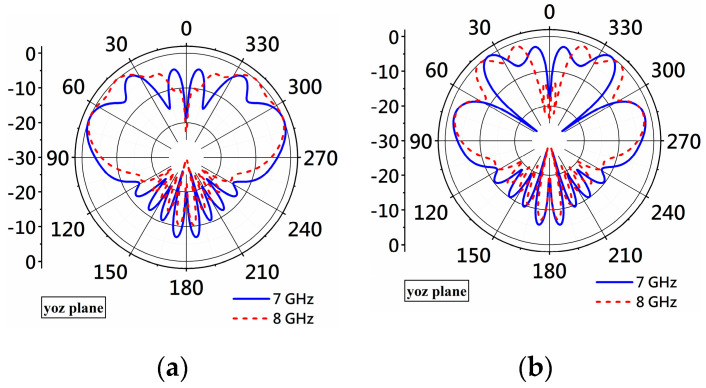
Simulated radiation patterns of Antenna II and Antenna III at 7 and 8 GHz on the yoz-plane. (**a**) Patterns of Antenna II. (**b**) Patterns of Antenna III.

**Figure 9 micromachines-14-02218-f009:**
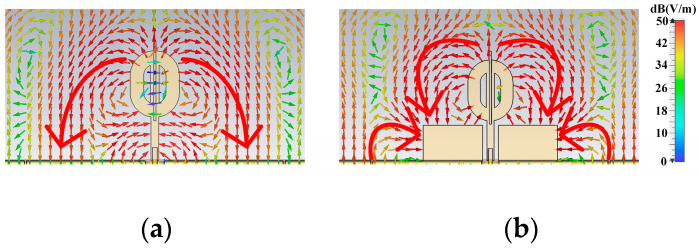
Side view of the electrical field distributions of Antenna II and Antenna III. (**a**) Field distribution of Antenna II. (**b**) Field distribution of Antenna III.

**Figure 10 micromachines-14-02218-f010:**
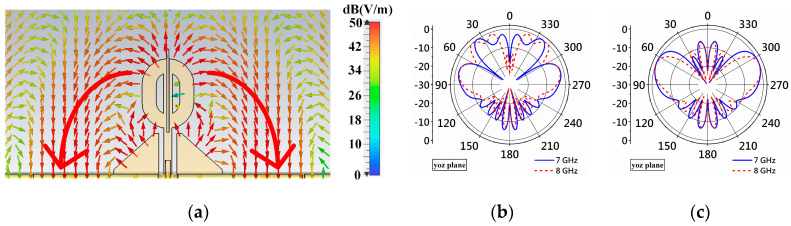
Field distribution of Antenna IV and relating radiation patterns of Antenna III and Antenna IV. (**a**) Electrical field distribution, (**b**) patterns of Antenna III, and (**c**) patterns of Antenna IV.

**Figure 11 micromachines-14-02218-f011:**

Three open-stub configurations, corresponding radiation patterns, and field distributions at 7 and 8 GHz. (**a**) Type 1. (**b**) Type 2. (**c**) Type 3. (**d**) Radiation pattern of type 1. (**e**) Radiation pattern of type 2. (**f**) Radiation pattern of type 3. (**g**) Electrical distribution of type 1. (**h**) Electrical distribution of type 2. (**i**) Electrical distribution of type 3.

**Figure 12 micromachines-14-02218-f012:**
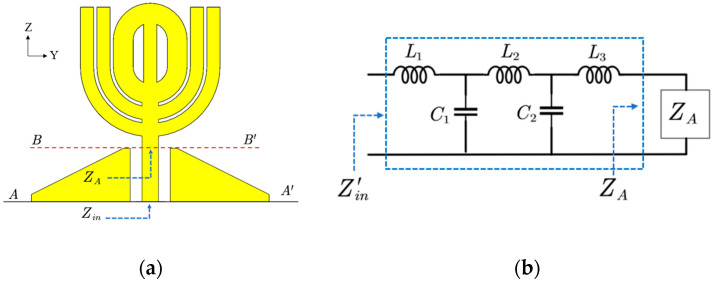
The proposed antenna, its equivalent circuit model, and corresponding reflection coefficients. The elements’ values of the equivalent circuit are *L*_1_ = 0.411 nH, *L*_2_ = 1.134 nH, *L*_3_ = 0.480 nH, *C*_1_ = 0.390 pF, and *C*_2_ = 0.359 pF. (**a**) Side view of the antenna. (**b**) Equivalent circuit. (**c**) Reflection coefficients.

**Figure 13 micromachines-14-02218-f013:**
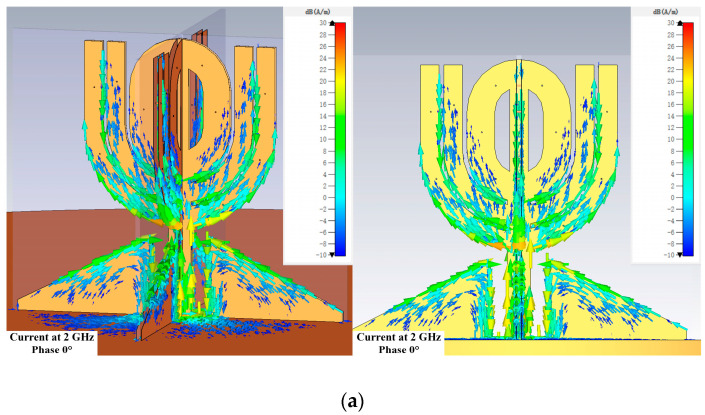
Current distributions at 2 GHz. The scale is from −10 dB(A/m) to 30 dB(A/m) in all the subfigures. (**a**) Perspective and side view when the phase is 0 degree. (**b**) Perspective and side view when the phase is 45 degrees. (**c**) Perspective and side view when the phase is 90 degrees. (**d**) Perspective and side view when the phase is 135 degrees.

**Figure 14 micromachines-14-02218-f014:**
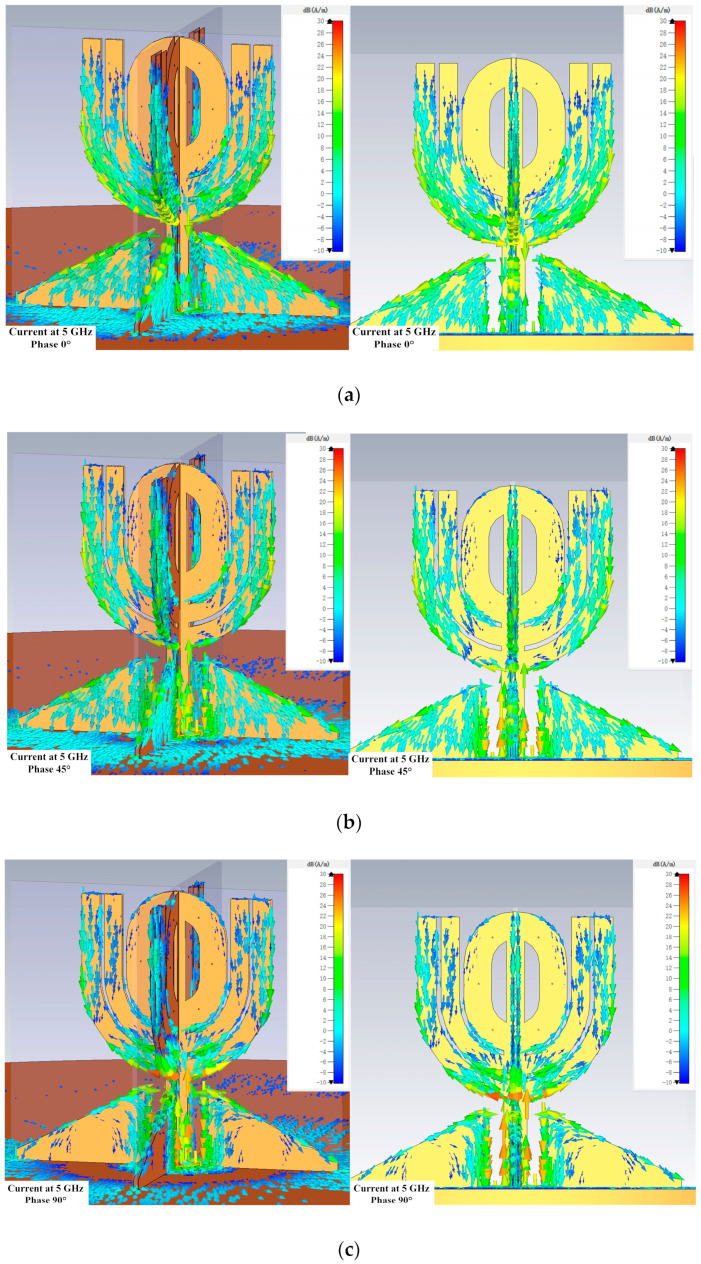
Current distributions at 5 GHz. The scale is from −10 dB(A/m) to 30 dB(A/m) in all the subfigures. (**a**) Perspective and side view when the phase is 0 degree. (**b**) Perspective and side view when the phase is 45 degrees. (**c**) Perspective and side view when the phase is 90 degrees. (**d**) Perspective and side view when the phase is 135 degrees.

**Figure 15 micromachines-14-02218-f015:**
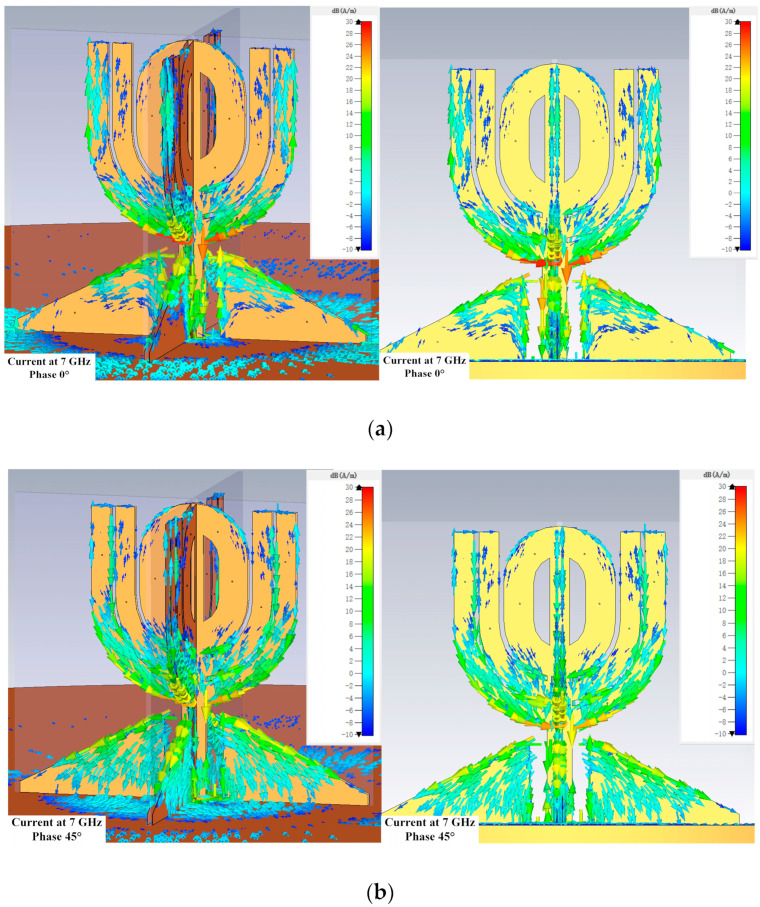
Current distributions at 7 GHz. The scale is from −10 dB(A/m) to 30 dB(A/m) in all the subfigures. (**a**) Perspective and side view when the phase is 0 degree. (**b**) Perspective and side view when the phase is 45 degrees. (**c**) Perspective and side view when the phase is 90 degrees. (**d**) Perspective and side view when the phase is 135 degrees.

**Figure 16 micromachines-14-02218-f016:**
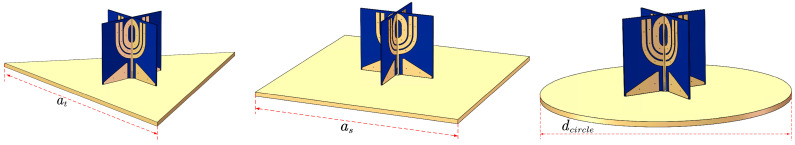
Antennas with equilateral triangular ground plane (**left**), square ground plane (**middle**), and circular ground plane (**right**).

**Figure 17 micromachines-14-02218-f017:**
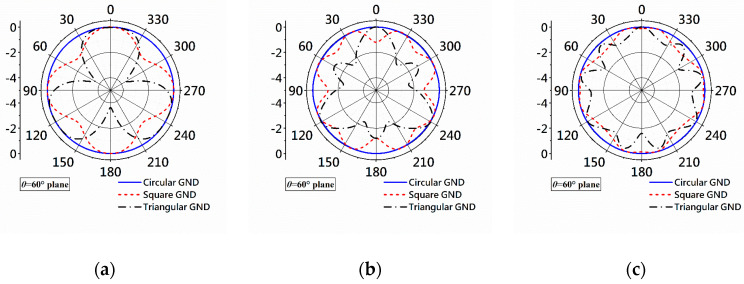
Radiation patterns obtained with triangular, square, and circular ground planes. (**a**) At 2 GHz. (**b**) At 5 GHz. (**c**) At 7 GHz.

**Figure 18 micromachines-14-02218-f018:**
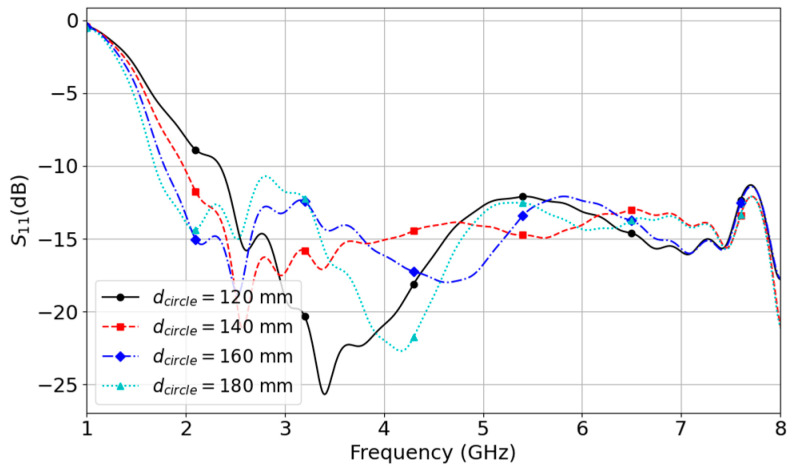
Reflection curves obtained with different diameters of the circular ground plane.

**Figure 19 micromachines-14-02218-f019:**
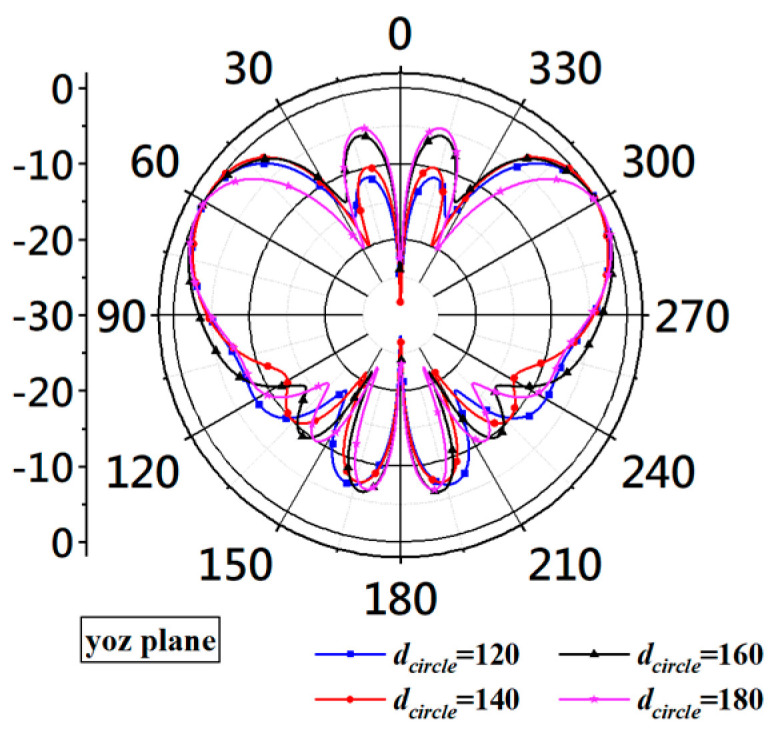
Radiation patterns at 5 GHz obtained with different diameters of the circular ground plane.

**Figure 20 micromachines-14-02218-f020:**
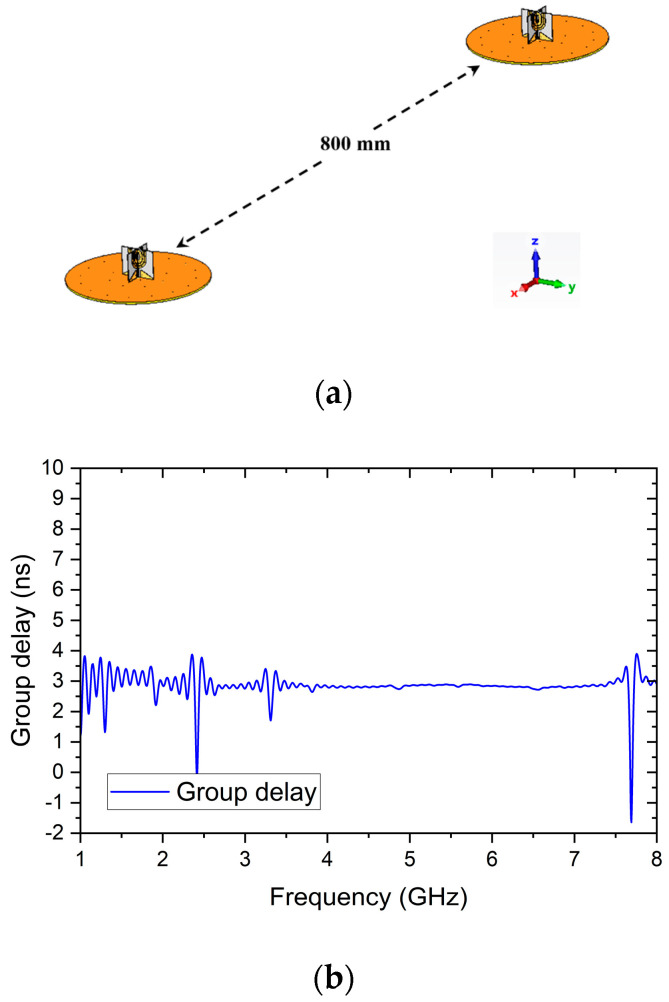
Time domain analysis configuration and result. (**a**) Simulation configuration. (**b**) Simulation result.

**Figure 21 micromachines-14-02218-f021:**
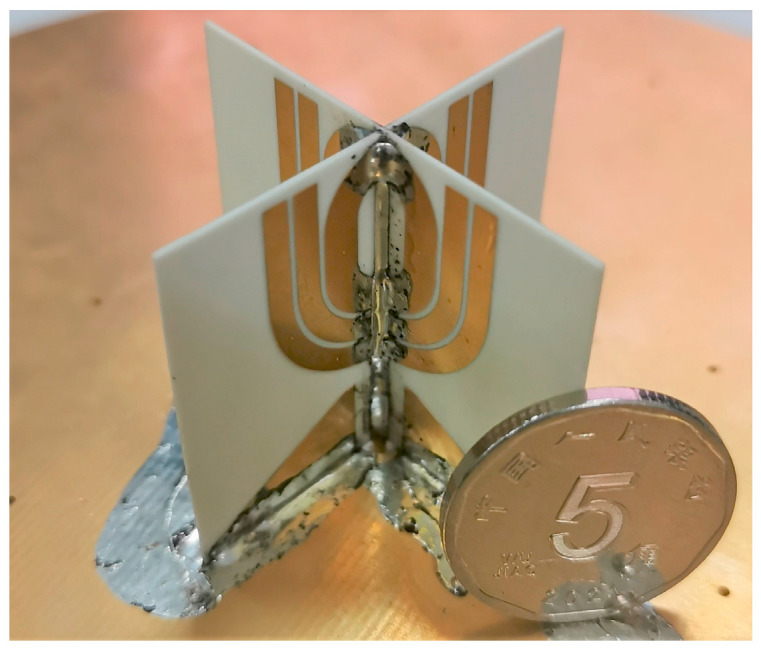
Photograph of the fabricated antenna.

**Figure 22 micromachines-14-02218-f022:**
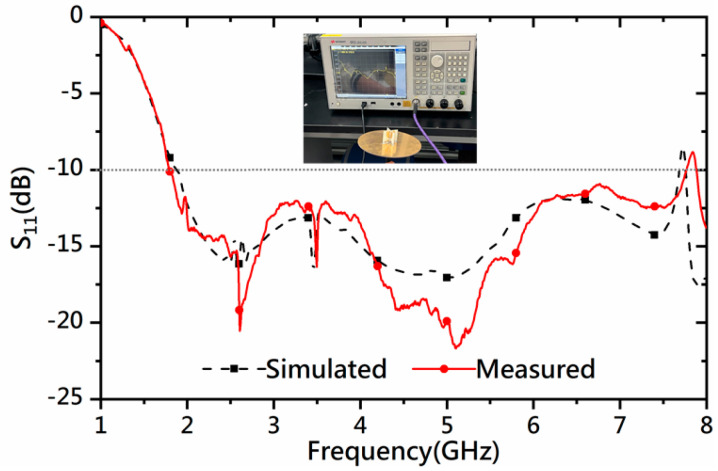
Simulated and measured *S*-parameters of the proposed antenna.

**Figure 23 micromachines-14-02218-f023:**
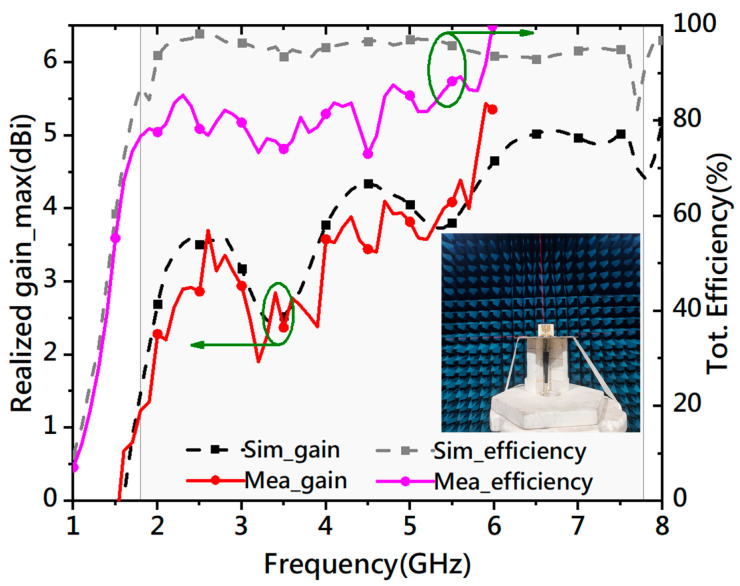
Simulated and measured maximum realized gains and total efficiencies of the proposed antenna.

**Figure 24 micromachines-14-02218-f024:**
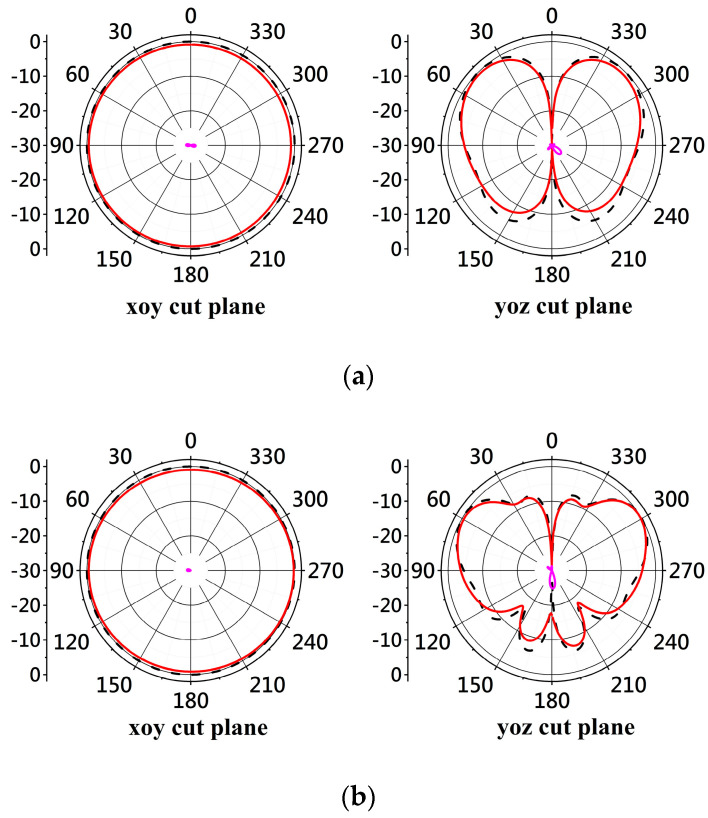
Simulated and measured radiation patterns at (**a**) 2 GHz. (**b**) 4 GHz. (**c**) 5.5 GHz.

**Table 1 micromachines-14-02218-t001:** The calculated dimensions of the proposed antenna.

Parameter	Value (mm)	Parameter	Value (mm)
*L*	37.2	*W* _3_	1
*H* _1_	30.5	*W* _4_	2.5
*H* _2_	8.1	*W* _5_	2
*H* _3_	1.1	*W* _6_	2
*H* _4_	9	*W* _7_	2
*H* _5_	15	*W* _8_	3
*H* _6_	4	*W* _9_	0.5
*W* _1_	15	*W* _10_	0.5
*W* _2_	1.8	*R*	5.2

**Table 2 micromachines-14-02218-t002:** Antenna performance comparison with previously reported antennas.

Ref.	Structure Type	Complexity (Material)	Freq. Range (GHz)	Fractional Bandwidth ^1^	Gain (dBi)	Dimension ^2^ (λ*_low_* ^3^)	Azimuthal Gain Variation (dB)
[2]	Patch	Simple (PCB)	4.8–11.1	79.4%	2.2–8.4	*D =* 0.88 *H* = 0.089	2
[3]	Cylinder	Complex (Metal)	25.6–40	43.9	9.2	*D* = 4.42 *H* = 4.5	3
[5]	Cylinder	Simple (PCB)	1.35–2.54	61%	0.5–1.5	*D* = 0.252 *H* = 0.27	1.5
[8]	Patch	Simple (PCB)	11.6–18	41.6%	4.6 (peak)	*D* = 2.79 *H* = 0.081	3
[11]	Patch	Simple (PCB)	2.75–7.43	91.9%	1–3.8	*L* = 0.32 *W* = 0.14	7
[13]	Cylinder	Complex (Metal)	0.69–2.84	121.5%	3.8–7.5	*D* = 0.144 *H* = 0.09	8
[14]	Patch	Simple (PCB)	0.62–2.6	123%	3.64–6.42	*L* = 0.155 *W* = 0.143	9
[15]	Patch	Simple (PCB)	2.85–11.85	122.4%	−1.2–3.2	*L* = 0.266 *W* = 0.19	5
[16]	Strip Wire	Complex (Metal/Plastic)	0.22–0.95	127.27%	−2.4–2	*L* = 0.21 *W* = 0.07 *H* = 0.04	5
This work	Patch	Simple (PCB)	1.8–7.77	124.8%	1.24–5.44	*L* = 0.223 *W* = 0.223, *H* = 0.183	1

^1^ Reflection coefficient is S11 < −10dB. ^2^ D, L, W, and H are the diameter, length, width, and height, respectively, of the antennas and the dimension does not include ground plane size. ^3^ The wavelength at the lower frequency of 10 dB return loss bandwidth.

## Data Availability

Data are contained within the article.

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
