# Peer review of "Wideband Omnidirectional Antenna Featuring Small Azimuthal Gain Variation"

_micromachines, 2023, doi:10.3390/mi14122218_

Round 1
Reviewer 1 Report
Comments and Suggestions for Authors
1. How were the order of the equivalent circuit model and the parameters in equation (1) determined in the article? Please add some further relevant knowledge.
2. Lines 140 and 141 of the article show that Simulation shows the matching can be controlled by tuning the height and length of the rectangular patches and their distance to the feed line. However, the article does not provide any image data to support this. How do the height and length of the rectangular patches control the matching?
3. Line 155: Simulated field distribution and pattern in Fig. 8 verifies that cutting matching patches’ corner reduces the Side lobe levels.
Please check that the cited image number is correct. It should be Figure 9, right?
4. Figure 9(b) shows the state at 7 GHz and Figure 10(d)(e)(f) shows the state at 8 GHz. Is this convincing? The optimization in Figure 10 is for Figure 9 unless there is some misunderstanding. So, what does the direction map look like at 7GHz after the optimization?
5. Does the concept of the average group delay presented in line 206 make sense?
Comments on the Quality of English LanguageProper English polishing is required.
Reviewer 2 Report
Comments and Suggestions for Authors
This paper presents a novel wideband omnidirectional antenna design that achieves a 1-dB
gain variation across its azimuthal plane within a bandwidth of 1.8 GHz to 7.77 GHz. Author has validated the antenna structure into simulation by using CST electromagnetic solver and had done fabrication and test. In order to discuss more the obtained results, author has to explain the difference obtained between simulation and measurement for gain, efficiency and the reflexion coefficient. For substate author has to give more details about tangent loss, thickness and dielectric relative permittivity. Something that it isn’t clear for figure.3, author has mentioned three antennas configuration and in figure.2 we have five which configurations have been chosen in figure.3?
Comments on the Quality of English Languageenglish can be more improved
Reviewer 3 Report
Comments and Suggestions for Authors
The authors have presented a paper on Wideband Omnidirectional Antenna Featuring Small Azimuthal Gain Variation. Following are my suggestions:
1. Kindly add parametric study of 2-3 important parameters
2. Add surface current diagram of the antenna
3. Explain the effect of ground plane variation on the performance of the antenna
4. Why circular ground plane was selected. What is the effect on the performance of the antenna if square, triangle or other shape of ground plane is selected.
5. The dimension of the ground plane should be added
Round 2
Reviewer 3 Report
Comments and Suggestions for Authors
The authors have answered all the queries and hence my suggestion is to accept the manuscript